# Generalizable and automated classification of TNM stage from pathology reports with external validation

Jenna Kefeli[1], Jacob Berkowitz[2,3,5], Jose M. Acitores Cortina [2,3,5], Kevin K. Tsang[2,3,5] & Nicholas P. Tatonetti [1,2,3,4]

Cancer staging is an essential clinical attribute informing patient prognosis and clinical trial eligibility. However, it is not routinely recorded in structured electronic health records. Here, we present BB-TEN: Big Bird – TNM staging Extracted from Notes, a generalizable method for the automated classification of TNM stage directly from pathology report text. We train a BERT-based model using publicly available pathology reports across approximately 7000 patients and 23 cancer types. We explore the use of different model types, with differing input sizes, parameters, and model architectures. Our final model goes beyond term-extraction, inferring TNM stage from context when it is not included in the report text explicitly. As external validation, we test our model on almost 8000 pathology reports from Columbia University Medical Center, finding that our trained model achieved an AU-ROC of 0.815–0.942. This suggests that our model can be applied broadly to other institutions without additional institution-specific fine-tuning.

Cancer stage, an important diagnostic and prognostic clinical attribute, is frequently used to identify patients for clinical trial recruitment and research cohort construction. While not routinely captured in the electronic health record, stage information can be found in patient pathology reports. Tumor registries, tasked with manually identifying stage from clinical notes and pathology reports, can take up to 6 months from diagnosis to extraction, at which point the opportunity for clinical trials or other treatments may have passed[1,2]. A shortage of cancer registry specialists suggests that this lead time may become even longer[3]. In this study, we present BB-TEN: Big Bird–TNM staging Extracted from Notes, a transformer-based method for the automated classification of TNM stage from pathology report text across 23 cancer types. Transformer-based methods have been applied to other clinical text[4], but have not been widely applied to pathology reports. We demonstrate that BB-TEN is generalizable to an independent institution, suggesting other institutions can use our method in an off-the-shelf capacity.

Extraction of cancer stage has been an ongoing effort. Previous studies have focused on single cancer types[5–8], used smaller-sized training[5,6,9] or testing[5,6,9,10] datasets (<1000 reports), and have relied on single-institution data without external validation and without proven generalizability[6,10]. In comparison, our work was performed on an initial pan-cancer dataset with approximately 7000 reports and then shown to extend in a generalizable fashion to an external pan-cancer dataset of almost 8,000 reports. Some studies required additional data beyond pathology report text as model input[5,6,10,11]. For ease-of-use, BB-TEN only requires the pathology report text as input and does not necessitate the inclusion of any other patient data types. In terms of methods, two studies employed older NLP methods (regular expression and customized rule-based approaches)[6,10], one utilized traditional machine learning methods[5], and another used a hybrid transformer-embedding and deep learning model[11]. In comparison, our method uses a recently-developed long-input transformer that directly ingests clinical-length pathology reports and fully incor-

[1]Department of Systems Biology, Columbia University, New York, NY, USA. [2]Department of Computational Biomedicine, Cedars-Sinai Medical Center, Los Angeles, CA, USA. [3]Cedars-Sinai Cancer, Cedars-Sinai Medical Center, Los Angeles, CA, USA. [4]Department of Biomedical Informatics, Columbia University, New York, NY, USA. [5]These authors contributed equally: Jacob Berkowitz, Jose M. Acitores Cortina, Kevin K. Tsang. e-mail: nicholas.tatonetti@cshs.org

porates long-range dependencies between tokens in model training. Further, these studies have not made their models publicly available, whereas we are releasing our trained TNM models to be directly utilized by other institutions. Finally, the use of state-of-the-art generative Large Language Models (LLMs), like OpenAI's GPT4, has been explored for prompt-based extraction of staging information from pathology reports[7–9,12]. These large models offer advantages over smaller transformer models in that they may require fewer training examples. However, the costs are significant, either in API access charges or in the hardware they require to run locally, their use may not be suitable for sensitive health information, and may be prone to hallucinations[13]. In contrast, we show that smaller transformer models, like BERT, achieve superior performance with fewer resources and no requirement to use third-party APIs.

The majority of prior work classified patients into broad TNM categories that do not cover the entire spectrum of clinical values[5,6,11], whereas in this study we classify reports specifically into *clinically relevant* TNM categorizations. Each of the possible category values was tailored to conform with current clinical usage and optimize for downstream utility. For example, we predict the full range of $N$ (0–3) vs. binary $N$ (0–1) because there are major distinctions in prognosis, suggested treatment, and research cohort selection across different N-values for different cancer types. In addition, granularizing $N$ is a substantially more difficult classification task. Other research studies[5,6,10,11] achieved high AU-ROC for prediction of binary $N$ (0–1), as did we in preliminary work; we ultimately selected (0–3) because (0–1) is a crude approximation for staging and not sufficient for a clinically useful end-model.

Similarly, prioritizing clinical relevance, we predict the full clinically actionable range of $M$, which is (0–1), as compared to resource[11] which predicts $M$ as (0–1, $X$). In preliminary work, we achieved high AU-ROC for $M$ (0–1, $X$); however, we removed $X$ as a possible prediction value to follow official AJCC guidelines, which call for the removal of $X$ from the pathologic staging vocabulary because $X$ is a non-clinically

actionable category [https://www.facs.org/media/j30havyf/ajcc_7thed_cancer_staging_manual.pdf]. Overall, our prediction categories and model output are more clinically relevant in light of current medical vocabulary and conform more closely with AJCC guidelines, particularly as compared to Preston, S. et al.[11].

In this study, we utilize recent advances in natural language processing to classify cancer stage directly from pathology report text[14]. We specifically use a new variant of the large language model BERT[15,16], which has a larger input capacity than previous versions, and show that our model performs better than standard BERT models. We also compare to a state-of-the-art LLM, Llama 3, and show that the BERT-based model performs better than the base Llama 3 model and better than the fine-tuned Llama 3 model in two out of three tasks with faster training time and fewer computational resources. To our knowledge, this is the first application of high-input capacity LLMs, i.e., LLMs for which long-input can be directly inputted without further modification, to pathology report text for classification of any prediction target.

## Results

Our overall approach consisted of (1) training a model using publicly available pathology reports and then (2) applying our trained model to a set of independent reports for validation of generalizability (Fig. 1A). Cancer stage consists of three elements: tumor size ($T$), regional lymph node involvement ($N$), and distant metastasis ($M$). Our prediction task consisted of classifying reports into TNM staging categories, with a separate model trained for each variable.

First, we selected 9523 pathology reports from The Cancer Genome Atlas (TCGA)[14,17]. The availability of TNM annotation in the TCGA metadata varied: 6887 reports were documented with known tumor size ($T$), 5678 reports with known regional lymph node involvement ($N$), and 4608 reports with known metastasis ($M$). All TNM values were based on pathologist review of relevant histology slides. Patients with known $T$ and $N$ values spanned 23 different cancer types, while patients with known $M$ status corresponded to 21 cancer types. Due to the large

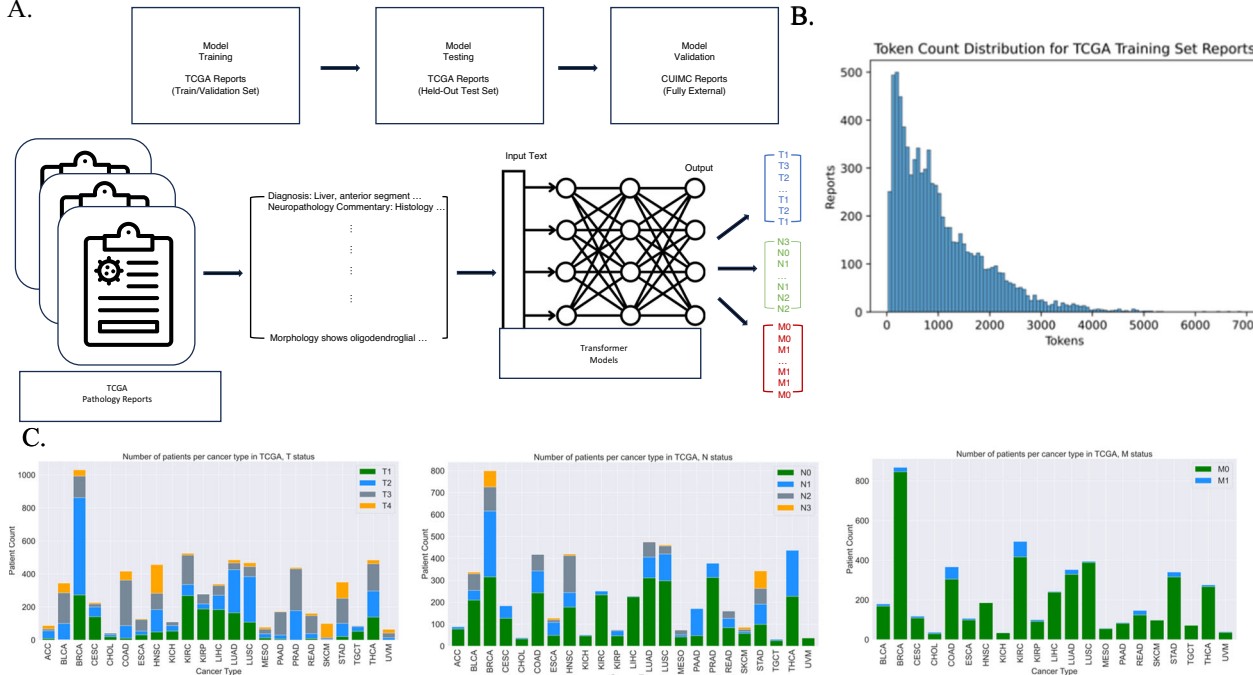

**Fig. 1 | Methodological overview and database summary statistics. A** Depiction of overall method. Top: Dataset separation into training/validation and held-out test sets (TCGA), as well as external validation (CUIMC). Bottom: Example TCGA pathology reports, inputted into separate transformer models to for TNM stage prediction. **B** Token distribution for TCGA training set reports. The ClinicalBERT

(CB) tokenizer was used to tokenize reports into pre-defined CB vocabulary. **C** Per-class distribution of TCGA pathology reports with TNM staging annotation. The distribution of TNM values varied substantially between cancer types. x-axis labeled as TCGA cancer-type abbreviations.

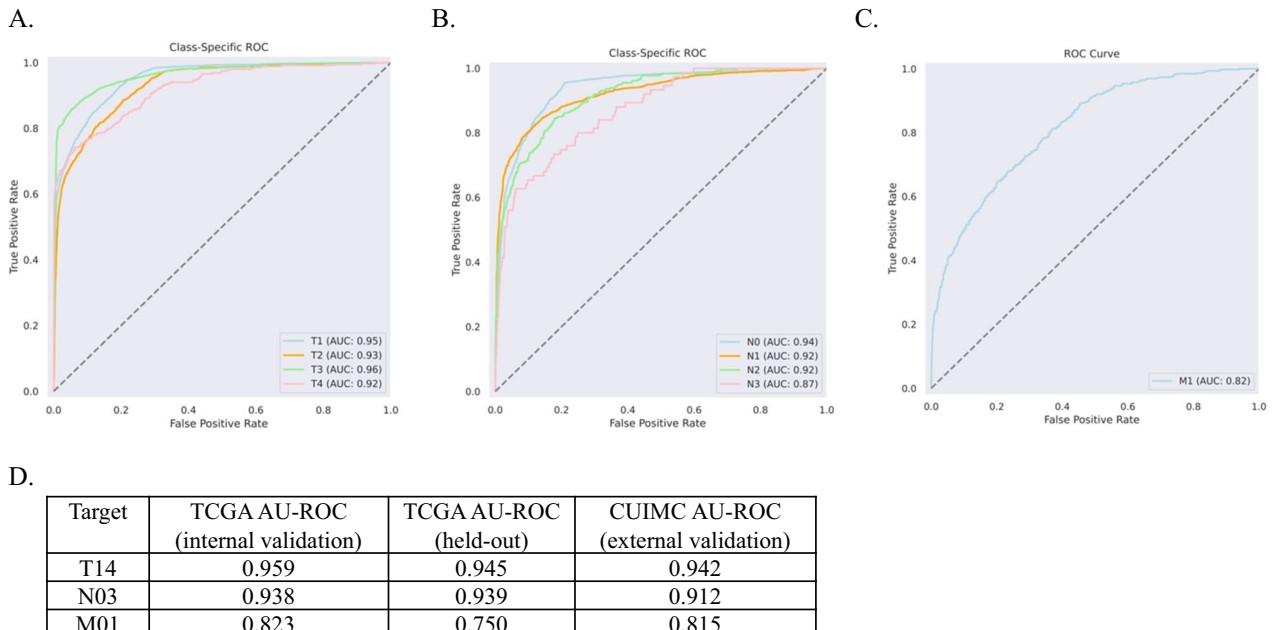

D.

| Target | TCGA AU-ROC (internal validation) | TCGA AU-ROC (held-out) | CUIMC AU-ROC (external validation) |
|---|---|---|---|
| T14 | 0.959 | 0.945 | 0.942 |
| N03 | 0.938 | 0.939 | 0.912 |
| M01 | 0.823 | 0.750 | 0.815 |

**Fig. 2 | Performance of final models. A–C** External validation using CUIMC data. Per-Class ROC Curves for each model: Tumor size (T14), Regional lymph node involvement (N03), and Distant metastasis (M01). **D** Best-performing models applied to TCGA held-out and CUIMC pathology reports. Models were selected based on TCGA internal validation set performance.

number of contributing pathologists and institutions represented in the TCGA dataset (Fig. S1), we observed that report structure, composition, and terminology varied greatly, even amongst single cancer-type subsets.

The complexity and size of the dataset suggested that a LLM would be appropriate, and that such a model should be generalizable once trained. To this end, we tested three pre-trained LLMs for our classification task[16,18,19] (Methods). Two models were pre-trained on a large set of publicly available clinical notes[20]. The first, ClinicalBERT (CB)[18], has been widely used in the field of clinical natural language processing, but is limited in both training and application by its maximum input capacity (512 tokens per document). Indeed, we found that over 66% of TCGA reports are greater than 512 tokens in length (Fig. 1B, Table S1). The second model, Clinical-BigBird (CBB)[16], recently released, has a vastly increased document input capability (4096 tokens per document) with proportionally fewer model parameters. The third model we evaluated was the most recent LLM release from Meta, Llama 3, which has state-of-the-art performance on common benchmarks[21]. We explored both the base model (zero-shot) and a fine-tuned version.

Classification was divided into individual tasks due to differences in patient training set size based on TNM status. Each classification target was assigned different integer-value ranges based on standard clinical use: $T$-values were in [1, 2, 3, 4], $N$-values were in [0, 1, 2, 3], and $M$-values were binary, [0, 1]. We denote these ranges as T14, N03, and M01 (Fig. 1C). We divided TCGA patients into training, validation, and held-out test sets. We selected the best-performing BERT-based model for each target based on validation set optimization. The model type that performed best across all three targets was CBB. We varied input size, finding that CBB models parameterized with larger input sizes generally performed better–CBB with 2048 input tokens performed best for T14 and N03 targets, and CBB with 1024 input tokens for the M01 target (Table S2). Validation set performance ranged from 0.82–0.96 AU-ROC. We then evaluated best-model performance on TCGA held-out test sets, with AU-ROC ranging from 0.75–0.95 (Figs. S2 and 3). The TCGA test sets were completely independent of the training dataset for each target. For comparison, we repeated this

analysis to evaluate the performance of Llama 3. We found the base model achieved F1s of 0.70, 0.84, and 0.81 for T14, N03, and M01, respectively. We then fine-tuned Llama 3 using the same experimental set up as the BERT-based models and the model achieved F1s of 0.78, 0.88, and 0.92, for T14, N03, and M01, respectively. BB-TEN outperformed the base Llama 3 for all three classification tasks and outperformed the fine-tuned Llama 3 (Llama3-FT) model for T14 and M01. We also performed a running time analysis and found that CB, BB-TEN (CBB), and Llama3-FT, took 3 min, 7.2 min, and 64.8 min, to train one epoch, respectively (Table S6).

We further evaluated our best-performing models on an independent set of pathology reports. We selected all pathology reports from Columbia University Irving Medical Center (CUIMC) from 2010–2019, and matched with tumor registry TNM annotation based on report date and diagnosis (see section "Methods"). As in the TCGA dataset, there was uneven coverage of TNM annotation across patients: 7792 patients corresponded to known $T$ status, 6140 patients corresponded to known $N$ status, and 2245 patients corresponded to known $M$ status (Table S3). Patients with $T$ status spanned 42 primary cancer sites; patients with $N$ status spanned 41 primary cancer sites, and patients with $M$ status spanned 40 primary cancer sites (Fig. S4).

The final models were not fine-tuned on CUIMC reports, but rather applied directly in an off-the-shelf capacity. We found that BB-TEN performed well, with AU-ROC ranging from 0.815–0.942 (Fig. 2A–D). For the multi-class targets T14 and N03, we found that per-class performance was consistently high (Fig. 2A, B). To ascertain whether the use of the CBB model type (with its increased complexity and much larger input size) made a difference in application to CUIMC data, we compared the best-performing CB model (as determined on the TCGA validation set) to the best-performing CBB model for the N03 task. We found that the best CB model performed at AU-ROC of 0.779, whereas the best CBB model produced AU-ROC 0.912 on CUIMC data (Table S4).

Finally, we tested whether including protected health information (PHI), such as name, date of birth, MRN, and gender, would have any impact on model performance for T14. We found that the best-performing model had a very slight improvement in performance

(difference in AU-ROC of 0.0001) when PHI was removed (Table S5). Given the small difference in performance, we do not consider PHI exclusion to be a necessary requirement for the use or application of our models.

## Discussion

Automating the classification of cancer stage from pathology reports would result in shorter turn-around time for clinical trial patient selection and research cohort construction. This would allow more patients to be routinely surveyed for inclusion in clinical trials and research studies. Automated stage annotation may be utilized in facilitating automated pathology report review for clinical use and prognostication, and in developing multi-modal models that combine both image and text, particularly as medical centers are increasingly digitizing pathology slides[22].

In this study, we applied a recently released transformer model, with higher input capacity than the traditional BERT model, to achieve consistently high predictive performance across multiple independent datasets. Importantly, we have made our models publicly available. Unlike other studies in clinical NLP, our models were trained on fully de-identified pathology reports, and therefore do not necessitate specialized approaches, such as differential privacy or federated learning, for their dissemination. Further, our models were trained on a highly diverse dataset spanning over 500 tissue source sites (Fig. S1), whereas previous studies have utilized data from a much narrower variety of sources – for example, one study only drew from 28 sites[11]. As we have demonstrated, the TNM models are consistently performant across institutions. We found that our TCGA-trained models performed as well on the CUIMC report set as on the TCGA held-out test set (Fig. 2D), suggesting that they are indeed generalizable. Many CUIMC reports did not contain explicit staging terms (15.3% of T14, 20.3% of N03, and 94.7% of M01 reports), suggesting that the models exhibit increased performance by going beyond direct extraction, inferring stage classification from context. Finally, we compared our BERT-based models to more recent state-of-the-art generative LLMs. Specifically, we evaluated the performance of Meta's newest release, Llama 3, and discovered that our BERT-based models performed better and faster. While the generative models provide generally strong performance across a wide range of NLP benchmarks, our results indicate that BERT-based models can be quickly and effectively tuned for specific classification tasks. This makes these models especially suitable for use with sensitive healthcare data, where data must be kept inside the institution, and across a wide range of computational infrastructure. Future work should focus on additional testing at external institutions to further validate the generalizability of our TNM models.

Although promising, our method has a number of notable limitations. Firstly, we abstracted the values of T, N, and M, converting for example T1a to T1. Re-training our models on a larger set of pathology reports would allow for a more detailed prediction target. Second, we were limited by our computational memory allotment, restricting the number of tokens per report to 2,048. If computational capacity were increased, one may extend to 4,096 tokens in order to capture longer reports at full length. Third, we developed our models based on current AJCC[23] definitions of stage; our model would need to be re-trained if AJCC definitions were substantially updated.

In addition, the M01 model did not perform as well as the T14 and N03 models overall. This is likely due to a number of factors inherent to the TCGA training data: (1) M01 was a particularly imbalanced dataset, with only 6.7% of reports having M1 annotation (Fig. 1C) due to TCGA design preference for non-metastatic cases, (2) many reports did not contain M0 or M1 explicitly, as compared to the other targets (95.1% of reports did not contain explicit M01, as compared to 66.5% for N03 and 35.8% for T14), and (3) TCGA annotations for M01 were at times inconsistent with report text (see section "Methods"). The M01

model is limited by the quality of the input data on which it is trained. In future work, the M01 model may be improved by training on a dataset with a great number of M1-annotated pathology reports and more consistent ground truth annotation.

## Methods

The research performed complies with all relevant ethical regulations; the institutional and IRB that approved the study protocol are Columbia IRB number AAAL0601. IRB waived informed consent for the study due to its retrospective and anonymized nature, minimal risk and lack of patient contact.

### TCGA pathology report dataset construction with TNM annotation

Pathology reports and associated TNM clinical metadata were downloaded from the TCGA Genomic Data Commons (GDC) data portal from https://portal.gdc.cancer.gov. Reports were initially stored in PDF format; in previous work, we converted the TCGA pathology report corpus to machine-readable plain text using OCR, performed extensive curation, and fully characterized the final TCGA report set. The final dataset spanned 9,523 reports, with 1:1 patient:report ratio[14].

TNM staging annotation was contained within the clinical metadata provided by TCGA. The TNM staging attribute used in this study is pathological stage, i.e., stage based on pathologist assessment of patient tumor slide(s) combined with previous clinical results. This value is distinct from clinical stage as provided by TCGA; we chose pathological rather than clinical staging for ground truth as (1) it is considered the diagnostic gold standard during the course of patient care and (2) information concerning pathologic staging is contained within report text. Staging was determined in a systematic manner by TCGA across all patients[17]. All data used for ground truth labeling was derived from the TCGA metadata as provided by the TCGA data portal.

TNM values were abstracted to numerical values, without additional letter suffixes–For example, N1B was converted to N1. Data availability, or TNM coverage, varied. A given report may have had no associated TNM data, full associated TNM data, or some combination of associated TNM values. Due to the difference in coverage, we separated the data by TNM data availability for individual classification tasks. Each target dataset consisted of non-uniform target value distributions, as displayed in Fig. 1C, to varying degrees.

Finally, TCGA annotation of M01 was found to be inconsistent. We examined a random sample of 10 pathology reports, with 5 reports annotated as M0 and 5 reports annotated as M1 in the TCGA metadata. We found that 5/5 reports annotated as M0 were labeled consistently with the AJCC definition of M0. However, we found that 2/5 reports annotated as M1 were not labeled consistently with the AJCC definition of M1 (distant metastasis), but rather contained characteristics similar to the reports labeled M0. From this, we observe that the ground truth labels for the M01 target may not be uniformly accurate, as they were found to be at times inconsistent with the AJCC definitions of distant metastasis and inconsistently applied among reports.

### Comparison of clinically pre-trained BERT-based models

For each target, we performed fine-tuning experiments using two model-types, CB[18] and CBB[16]. Both models had been pre-trained on a set of clinical notes (MIMIC III[20]). CB has consistently performed at a high level across a variety of clinical natural language processing tasks[24–26]. Model CB contains 108.3 M parameters and is based on the classic BERT architecture[15]. CB is, however, very limited by a maximum input document length of 512 tokens. As a result, reports longer than 512 tokens are truncated during training, and text beyond 512 tokens is not used for model learning. In addition, when applying the model to an external dataset, reports must again be truncated to 512 tokens, so that any information contained within text beyond 512 tokens is not

applied toward model prediction. As many real-world reports are longer than 512 tokens, this is a serious limitation.

A more recent model, CBB, has 128.1 M parameters and adopts the computationally-optimized *BigBird* architecture[27]. Bigbird is based on the BERT architecture but differs in the specification of the attention mechanism. Briefly, a sparse attention mechanism allows for longer inputs to be computationally tractable, providing linear run-time with number of input tokens (compared to the quadratic run-time of BERT) and better performance on benchmark tasks[27]. As a result, model CBB has a vastly increased document length capacity (4096 tokens), which allows the use of entire-length reports in both training and application. For example, in the TCGA pathology report dataset, over 66% of reports in the TCGA dataset contain >512 tokens (Table S1), while 12.9% have report length >2048 tokens, and only 0.7% have reports >4096 tokens.

## Multi-class classification tasks utilizing the TCGA pathology report dataset

We separated reports into reports with M01 annotation, reports with N03 annotation, and reports with T14 annotation. M01 annotation had the least coverage in the TCGA dataset overall. Each report set was divided into training (70%), validation (15%), and held-out test (15%) sets. As each patient corresponded to a single report, no patient spanned more than one train/validation/test (TVT) subset. In addition, when separating the reports into TVT subsets, we balanced on TNM value composition so that the same balance of values was consistent across TVT subsets. This allowed for fair comparison of performance across TVT subsets, with no TVT subset having a greater imbalance than the dataset overall.

Independent models were trained and hyperparameter-optimized for each of M01, N03, and T14 classification targets separately, as specified below. We evaluated model performance based on macro AU-ROC and per-class AU-ROC (in a one-vs-all capacity). Each target was evaluated separately.

## Hyperparameter optimization, model fine-tuning, and model selection

**ClinicalBERT and Clinical-BigBird.** For hyperparameter optimization, we performed an iterative grid search across two learning rates, three batch sizes, and three random seeds (used for train/validation split). Due to memory limitations, the maximum number of input tokens per document that we were able to implement was 2048 input tokens. We used 512 input tokens for CB (the maximum allowed by the CB model), but for CBB we experimented with 512, 1024, and 2048 (the maximum allowed by our hardware). We fine-tuned each model for 30 epochs. Run-time of CBB experiments was substantially longer than that of CB experiments, with 2048 input token CBB (CBB-2048) instantiations taking almost 24 h of training run-time per parameter combination. We evaluated model performance depending on TCGA validation set AU-ROC, selecting the best final model per target based on this metric. We found that CBB-2048 was the best model type for T14 and N03 targets, whereas CBB-1024 was the best for the M01 target (Table S2). The final TNM models are made publicly available through Huggingface (https://huggingface.co), which is a widely used Python library for publishing and downloading LLMs.

**Llama 3.** Llama 3, developed by Meta AI, is a large-scale language model with 8 billion parameters, designed to capture a wide range of general knowledge and demonstrate state-of-the-art performance on various natural language understanding benchmarks. To adapt Llama 3 for our specific clinical classification tasks, we employed the Low-Rank Adaptors (LoRa) methodology, which allows for efficient fine-tuning of large pre-trained models. LoRa introduces low-rank matrices to model's attention and feed-forward layers, enabling us to update only a small subset of the model's parameters while keeping the rest frozen.

This approach significantly reduces the computational resources required for fine-tuning and allows for rapid adaptation to new tasks. For the fine-tuning process, we initialized Llama 3 with its pre-trained weights and introduced LoRA adaptors with rank: $r = 16$ and scaling factor alpha = 16. We fine-tuned the model on the TCGA Pathology Report Dataset, targeting the classification layers for the M01, N03, and T14 staging annotations. The fine-tuning was conducted over 3 epochs with a batch size of 16 and a learning rate of 3e-4. We tested the fine-tuned model as well as the base model.

## Evaluation of training time

In order to compare the different models' training time, we set an experiment with the same conditions for the three models. We used one instance of NVIDIA A100 GPU. The specifications for this model include 80 GB of memory and 2 TB/s of memory bandwidth. The results can be seen in Table S6, showing the direct correlation between parameters and training time.

## Characterization of CUIMC pathology report dataset

We retrieved all reports from the CUIMC pathology report database, between 2010–2019. We removed empty reports and outside consultation reports. We selected for reports with the surgical pathology label, as this label indicated histopathology slide analysis in contrast to other report types generated by the pathology department. Report text remained intact, not pre-processed. TNM stage annotation data were located in a separate metadata table, derived from the tumor registry. We selected for patients with non-empty TNM values.

We employed three attributes to match report text to patient TNM annotation: patient ID, report date (matched to TNM diagnosis date), and TNM-primary site (Fig. S4). Patient ID was matched exactly across the two databases. For date-matching, we allowed up to 90 days between report date and diagnosis date, as there is a lead-time/delay between pathologist documentation and official tumor registry stage extraction. We observed that the number of reports overall, as well as the number of reports per patient, increased as the time-window was expanded from 0 to 90 days. Additionally, we observed that a single patient may have multiple pathology reports potentially associated with a given TNM annotation, within the same time-window. We therefore imposed an additional matching requirement to ensure report-annotation relevancy, selecting the most relevant report as that which has the greatest number of report string matches to the TNM-associated primary site value. At this stage, the vast majority of patients were associated with a single TNM-report match. However, in the event that multiple reports were equally relevant, we concatenated reports to ensure that all relevant TNM-information would be captured.

In the final CUIMC dataset, most reports had associated T14 annotation, and the least number of reports had M annotation, similar to the TCGA dataset (Table S3A). We tabulated the class imbalance for each target (Table S3D). We found that T4 and N3 are the least-prevalent classes per target, as was the case for the TCGA report set (Fig. 1C). We also found that the proportion of M1 is higher in the CUIMC dataset (20.1%) as compared to the TCGA dataset (6.7%). The range of diseases is larger for the CUIMC reports as compared to TCGA reports: The TCGA dataset ranged from 21–23 cancer types, whereas the CUIMC dataset spans 40–42 primary sites (although these terms are not directly comparable). We plotted the primary site distribution for each target report set (Fig. S4), finding that the distributions are similar across the three targets. As in the TCGA dataset, breast and lung are two of the most prevalent cancer sites, across all three targets. Finally, using the CBB tokenizer, we computed token statistics for each target dataset (Table S3C). Overall, we found that CUIMC pathology reports were longer than TCGA pathology reports, both on average and at maximum report length.

Additionally, we explored the demographics of our dataset and included them Table S3B.

## Application of TCGA-trained models to CUIMC dataset

TNM models were applied directly to the entire CUIMC report set (without any additional fine-tuning). As before, we calculated AU-ROC to evaluate model performance. We found that, as for TCGA validation and held-out test sets, M01 was the least well-performing model (as compared to the T14 and N03 models).

We compared the CUIMC performance of our TNM models to those of Abedian et al. [5,10], which was the most comparable to ours in terms of the use of pathology report text as sole input, the predicted TNM target value ranges (T14, N03, and M01), and the inclusion of multiple cancer types in both train and test sets. Abedian et al. reported F1, rather than AU-ROC. We computed F1 for our models and compared our results to the pan-cancer test set results in ref. 5 (Table S3E). We found that our T14 model performed on-par with[5], our N03 model performed somewhat better, and our M01 model performed substantially better than the equivalent model in ref. 5.

We performed three additional experiments to probe our external validation results. First, although we found that the CBB model-type achieved the best performance on the TCGA report set, we were interested in whether this result would hold for CUIMC reports. To test this, we applied the best-performing TCGA-trained CB model to the CUIMC report set to predict the N03 target. There was a large difference in performance across all evaluation metrics, including overall macro and per-class AU-ROC between CB and CBB (Table S4). CBB likely performed better than CB due to its increased complexity as well as its increased input token size (Table S3C).

Second, we tested whether our primary parameter for report-diagnosis matching, number of days between diagnosis and report date, had any impact on CUIMC performance. We ran the TCGA-trained models on CUIMC data for each target separately for 0, 10, and 30 days; we compared the results to the performance we achieved with 90-day report-matching (Fig. S5). In this sensitivity analysis, we found that AU-ROC remained stable as the number of days was varied. For the multi-class targets, T14 and N03, we plotted per-class changes over time, finding that there is a slight increase in per-class AU-ROC as the number of days increases. The magnitude of AU-ROC increase across number of days varies by class. The least-prevalent classes (e.g., T4 and N3) have the largest gain in AU-ROC as number of days increases; this is likely due to the increased likelihood of report relevance as number of days increases.

Finally, we tested the removal of PHI, such as medical record number, date of birth, etc., from the preamble of each report for the T14 target. In the CUIMC dataset, most of the patient-identifying text was located in the first few lines of each report (whereas diagnosis information was not typically contained in this preamble section). Our hypothesis was that the model may perform better without extraneous patient details, particularly as these types of details had not been seen by the model when trained on the de-identified TCGA report set. However, we observed only a 0.0001 AU-ROC gain when PHI was removed (Table S5). We determined that PHI removal was not necessary for external validation, as increased pre-processing effort would potentially lead to only a very small performance gain.

## Software requirements

For the training and testing of our model, we utilized the following Python (version 3.12) packages: numpy (version 1.26.4) for numerical computations, pandas (version 2.2.2) for data manipulation and analysis, scikit-learn (version 1.4.2) for machine learning algorithms, scipy (version 1.13.0) for scientific computing, seaborn (version 0.11.2) for data visualization, transformers (version 4.40.2) for natural language processing, and torch (version 2.3.0) for deep learning. Specifically, for the llama3 model, we employed accelerate (version 0.30.0) for optimizing training speed, bitsandbytes (version 0.43.1) for efficient computation, evaluate (version 0.4.2) for performance assessment, huggingface-hub (version 0.23.0) for model sharing, and peft (version 0.10.0) for parameter-efficient fine-tuning.

## Reporting summary

Further information on research design is available in the Nature Portfolio Reporting Summary linked to this article.

## Data availability

Data from the TCGA and the CUIMC EHR were used in this analysis. The data supporting the findings described in this manuscript are available in the article and in the Supplementary Information or from the corresponding author upon request. The TCGA pathology report text can be found at https://github.com/tatonetti-lab/tcga-path-reports under an MIT License. De-identified data from the CUIMC EHR (i.e., pathology reports) will be made available in a controlled access manner. Controlled access is required due to the sensitivity of the data in the pathology reports used in this study. Researchers who wish to access the data must be trained in and abide by HIPAA policies and may reach out to the corresponding author to initiate the data access request who will respond to each request within 30 days. Data access and use agreements will be determined by CUIMC according to institutional published guidelines.

## Code availability

Python scripts used in this study can be found on Github: https://github.com/tatonetti-lab/tnm-stage-classifier[28] Models generated by this study can be found on Huggingface: https://huggingface.co/jkefeli/CancerStage_Classifier_T https://huggingface.co/jkefeli/CancerStage_Classifier_N https://huggingface.co/jkefeli/CancerStage_Classifier_M.

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

## Acknowledgements
We would like to acknowledge the help of Benjamin May for providing pathology reports from the CUIMC data warehouse. J.K. and N.P.T. are supported by NIH NIGMS R35GM131905.

## Author contributions
J.K. conceived the study, performed data analysis, and wrote the manuscript. J.B., J.A.C. and K.K.T. performed data analysis and wrote/edited the manuscript. N.P.T. supervised the study, directed study analysis, and wrote/edited the manuscript.

## Competing interests
The authors declare no competing interests.
