## [Peer Review File · Nature Communications]

Reviewers' Comments:

Reviewer #1:

Remarks to the Author:

The paper deals with training a BERT model on TCGFA data and testing on external data from Columbia University. Two models are in focus, namely ClinicalBERT and Clinical BigBird (CBB). The major argument for CBB is the much longer input length, i.e., 4096 tokens per document (pathology reports). Although the authors have not been able to use the full advantage due to memory shortage. The TNM classification (or staging) is used as the outcome with Tumour-values in [1, 2, 3, 4], Node-values in [0, 1, 2, 3], and Metastasis-values in [0, 1]. Validation set performance ranged from 0.823-0.959 AU-ROC.

Strength: Paper is very well written. Data is explained clearly. External validation is performed. Results appear on the higher end.

Weakness: No innovation is put forward. No extensive comparative study with other SOTA LLMs is performed.

Reviewer #2:

Remarks to the Author:

Dear Authors,

Thank you for sharing your work on predicting pathologic TNM staging from pathology reports. I am providing my comments below using recommendations from the journal.

- What are the noteworthy results?

It is noteworthy that the authors are planning to make their models publicly available.

- Will the work be of significance to the field and related fields? How does it compare to the established literature? If the work is not original, please provide relevant references. To the best of my understanding, the methodology discussed in this work was previously reported in a similar fashion in reference 8. The two stated differentiations are “granular, clinically relevant TNM categorizations” and “releasing our trained TNM models to be directly utilized by other institutions.” For the first stated differentiation, reference 8 also considers granular TNM categorizations with the exception of combining N1, N2, and N3.

- Does the work support the conclusions and claims, or is additional evidence needed?

The work reports its model performances.

- Are there any flaws in the data analysis, interpretation and conclusions? Do these prohibit publication or require revision?

One item that is unclear to me is whether the models are indeed targeting pathologic staging values and if so, how the pathologic staging values are gathered. In the text, “The TNM staging attribute used in this study is ‘pathological stage,’ i.e., stage based on pathologist assessment of patient tumor slide(s).” This misses the critical point that pathological staging has to do with a surgery event. According to AJCC, “Pathological staging can be determined when a patient has surgery to remove a tumor. Pathological staging combines the results of both the clinical staging with the surgical results.” It’s not clear to me if the authors are linking pathological staging to pathologist determined results. Clinical staging, on the other hand, is determined before surgery and can also involve pathology report (biopsy) and pathologist determination.

To accurately model the pathologic staging, one needs to filter out pathology reports prior to surgery so to not confuse pathologic and clinical staging. It is unclear how this was addressed in the paper.

- Is the methodology sound? Does the work meet the expected standards in your field? The methodology of applying a BERT-variant for classification is common practice in the field.

- Is there enough detail provided in the methods for the work to be reproduced?

There isn’t enough details provided for this work to be reproduced. For example, to reproduce the results, one would need learning rate, number of training epoch, and how the 9,523 patients were selected. It is also unclear how the one pathology report per patient was selected. It is not uncommon to have more than one pathology report per cancer patient.

Thank you for your re-consideration of our manuscript. We hope to show that our initial submission was not sufficiently clear regarding specific points mentioned by the reviewers as well as our improvement as compared with previous research.

Overall Response

Thank you for the reviews of our manuscript. As you noted the reviewers thought the impact and quality of the work was high. However, because of a lack of clarity in the manuscript it was viewed as not novel. The work is significantly different from previous work in several ways. While prior work was good as proof of concept, ours demonstrates the non-trivial innovations required to make the tool useful in practice for both research and clinical purposes.

Our work is superior to previous work because it is (1) publicly available and easy to apply, (2) contains clinically relevant TNM values, which is a non-trivial concern, (3) is pan-cancer applicable, and (4) has demonstrated generalizability. We have thought about this deeply from a clinical perspective. Specifically, N[0/1] (binary) classification as done in previous work vs. N[0-3] classification done in ours is a non-trivial leap. Granularity is important. In preliminary work to classify binary N[0/1], we achieved very high AU-ROC (as did Yala 2017, with simple machine learning methods, and Preston 2023, with deep learning methods). However, N[0/1] not sufficient for a clinically useful end-model, as it is a very crude approximation for staging. There are serious distinctions in prognosis, suggested treatment, and research cohort selection across different N values for different cancer types. We therefore expanded to the more difficult and clinically useful N[0-3] classification task. Similarly, we found that M[0/1/X] (Preston 2023) is less clinically relevant. We removed X (unknown) as a possible TNM value when we found that AJCC guidelines do not include X because it is clinically non-actionable.

Response to Reviewer 1:

Thank you for reviewing our manuscript. We appreciate the positive feedback and the concern regarding innovation. We have responded below to each point and expanded our manuscript so that it is clearer regarding the concerns mentioned in your review.

Reviewer Comment: “Strength: Paper is very well written. Data is explained clearly. External validation is performed. Results appear on the higher end.
Weakness: No innovation is put forward. No extensive comparative study with other SOTA LLMs is performed.”

SOTA LLMs: We compared 2 SOTA LLMs, ClinicalBERT and Clinical-BigBird, in our paper (please refer to the 3rd paragraph of Results and section in Methods entitled “Comparison of Clinically Pre-Trained BERT-Based Models”). We also

compared our model to a custom NLP approach introduced in reference 5 (Abedian 2021), which is the most recent and comparable study to ours in terms of input type and target classification. This allows for an apples-to-apples comparison. We found that our model performed consistently better (Table S3-D). Please see the 2nd paragraph of the “Application of TCGA-Trained Models to CUIMC Dataset” section in methods, which we include below for your reference:

We compared the CUIMC performance of our TNM models to those of Abedian et al. (2021) [5], which was the most comparable to ours in terms of the use of pathology report text as sole input, the predicted TNM target value ranges (T14, N03, and M01), and the inclusion of multiple cancer types in both train and test sets. Abedian et al. reported F1, rather than AU-ROC. We computed F1 for our models and compared our results to the pan-cancer test set results in [5] (Table S3D). We found that our T14 model performed on-par with [5], our N03 model performed somewhat better, and our M01 model performed substantially better than the equivalent model in [5].

Table S3. (D) Performance of TNM models on CUIMC data, as compared to TNM model performance in (Abedian et al., 2021), across all cancer types. For (Abedian et al., 2021) results, we selected the pan-cancer “random subtypes” test set; this matched most closely with the inclusion of all subtypes in the CUIMC dataset. F1 is micro-computed.

Target	F1 (Kefeli, 2023)	F1 (Abedian, 2021)
T14	0.78	0.78
N03	0.86	0.78
M01	0.82	0.11

Unfortunately, we were unable to directly compare our approach to the LLM introduced in [8] (Preston 2023) because that model is not publicly available due to PHI (protected health information) regulations. In addition, as the model from [8] has different input/output to ours – namely, it requires additional types of information in addition to pathology reports in order to run – any comparison would not be apples-to-apples.

Innovation: Please see comments above in the “Overall Response,” revisions to the manuscript, and the following response below. In comparing our manuscript to the study performed by reference 8 (Preston 2023), we would like to clarify that our underlying method is substantially different, our training data is much more diverse, and our model is truly available “off the shelf” (compared to the non-availability of the Preston 2023 model).

- **Difference in underlying model:** Our study utilizes a recently-developed long-input transformer that directly ingests clinical-length pathology reports and fully incorporates long-range dependencies between tokens in model training. In addition, model input (entire reports, no EHR component) and output (clinically-actionable targets) are substantially different as compared to any previous related studies.
- **Diversity of training data:** Our training data is more diverse. Preston (2023) use data from 28 cancer care sites from 5 states in their study; we used the TCGA dataset, which contains TNM annotation for >500 report-contributing sites nationally (**see Figure S1**). Due to this diversity, both training and testing (evaluation) are inherently more difficult, but the resultant model is expected to be more generalizable.
- **Utility - Public availability, access, and useability:** Our model is available and useable, whereas the model from Preston (2023) is not. Preston (2023) used multiple data sources (pathology reports + radiology assessments + clinical progress notes) and a large non-publicly available dataset - Their model would require significant computational resources with long run-time for training. However, their trained model is not available for use. In their paper, the authors claim that other researchers can simply re-do their process to derive a similar model for their own application and use. This is simply not practically nor logistically feasible. It is clear on the Github repository associated with Preston (2023) that the only available data consists of a skeleton example script to train your own model from scratch, with no model weights. Essentially, **there is no usable model provided to the end-user.** Their model is not publicly available, whereas ours is. This is crucial for utility.

The below paragraphs were added to the Introduction to clarify the points above:

Extraction of cancer stage has been an ongoing effort. Previous studies have focused on single cancer types [5-6], used smaller-sized training [5-6] or testing [5-7] data sets (<1,000 reports), and have relied on single-institution data without external validation and without proven generalizability [6-7]. In comparison, our work was performed on an initial pan-cancer dataset with approximately 7,000 reports and then shown to extend in a generalizable fashion to an external pan-cancer dataset of almost 8,000 reports. Some studies required additional data beyond pathology report text as model input [5, 8]. For ease-of-use, our model only requires the pathology report text as input, and does not necessitate the inclusion of any other patient data types. In terms of methods, two studies employed older NLP methods (regular expression and customized rule-based approaches) [6-7], one utilized traditional machine learning methods [5], and another used a hybrid transformer-embedding and deep learning model [8]. In comparison, our method uses a recently-developed long-input transformer that directly ingests clinical-length pathology reports and fully incorporates long-range dependencies between tokens in model training. In addition, none of the previous research studies have made their models publicly available, whereas we are releasing our trained TNM models to be directly utilized by other institutions.

The majority of prior work classified patients into broad TNM categories that do not cover the entire spectrum of clinical values [5-6, 8], whereas in this study we classify reports specifically into clinically relevant TNM categorizations. Each of the possible category values was tailored to conform with current clinical usage and optimize for downstream utility. For example, we predict the full range of N (0-3) vs. binary N (0-1) because there are major distinctions in prognosis, suggested treatment, and research cohort selection across different N values for different cancer types. In addition, granularizing N is a substantially more difficult classification task. References [5,8] achieved high AU-ROC for prediction of binary N (0-1), as did we in preliminary work; we ultimately selected (0-3) because (0-1) is a crude approximation for staging and not sufficient for a clinically useful end-model.

Similarly prioritizing clinical relevancy, we predict the full clinically actionable range of M, which is (0-1), as compared to resource [8] which predicts M as (0-1, X). In preliminary work, we achieved high AU-ROC for M (0-1, X); however, we removed X as a possible prediction value to follow official AJCC guidelines, which call for the removal of X from the pathologic staging vocabulary because X is a non-clinically-actionable category [https://www.facs.org/media/j30havyf/ajcc_7thed_cancer_staging_manual.pdf]. Overall, our prediction categories and model output are more clinically relevant in light of current medical vocabulary and conform more closely with AJCC guidelines, particularly as compared to [8].

The following was also added to the Discussion (paragraph 2):

Further, our models were trained on a highly diverse dataset spanning over 500 tissue source sites (Fig S1), whereas previous studies have utilized data from a much narrower variety of sources – for example, one study only drew from 28 sites [8].

Response to Reviewer 2:

Thank you for taking the time to review our manuscript. We have included comments below to address the items referenced in the review. The original review is included and our responses are indented. Thank you again for your re-consideration.

Reviewer Comment: “To the best of my understanding, **the methodology** discussed in this work was previously reported in a similar fashion in reference 8. The two stated differentiations are "granular, clinically relevant TNM categorizations" and "releasing our trained TNM models to be directly utilized by other institutions." For the first stated differentiation, reference 8 also considers granular TNM categorizations with the exception of combining N1,N2, and N3.”

Thank you for your comment. The methodology in our paper is substantially different from reference 8: The type of transformer model, the model input (entire reports, no EHR component), and the training corpus are substantially different.

Importantly, the prediction targets are also different. In terms of granularization, please see the revisions we made in the manuscript (paragraphs 3 and 4 of the Introduction). To clarify briefly, in our work we predict *clinically actionable targets*, as per the AJCC guidelines, including granularized N0-3 as well as M01 (without X). This makes our models more clinically relevant and more directly useable by institutions looking to automate staging extraction.

Reviewer Comment: “One item that is unclear to me is whether the models are indeed targeting pathologic staging values and if so, how the pathologic staging values are gathered. In the text, “The TNM staging attribute used in this study is ‘pathological stage,’ i.e., stage based on pathologist assessment of patient tumor slide(s).” This misses the critical point that pathological staging has to do with a surgery event. According to AJCC, “Pathological staging can be determined when a patient has surgery to remove a tumor. Pathological staging combines the results of both the clinical staging with the surgical results.” It’s not clear to me if the authors are linking pathological staging to pathologist determined results. Clinical staging, on the other hand, is determined before surgery and can also involve pathology report (biopsy) and pathologist determination. To accurately model the pathologic staging, one needs to filter out pathology reports prior to surgery so to not confuse pathologic and clinical staging. It is unclear how this was addressed in the paper.”

Thank you for your comment. The models are indeed targeting pathologic staging, not clinical staging, as described in the paper (Methods, 2nd paragraph). Pathologic staging was determined by TCGA in a systematic manner across all patients. All data used for ground truth labeling was derived from the TCGA metadata as provided by the TCGA data portal.

All of the TCGA labels that were used in this study were denoted as “pathologic staging,” as distinct from the “clinical staging” variable also present in the TCGA metadata. We decided to model “pathologic staging” rather than “clinical staging” because it is more accurate and closer to an ideal ground truth. Specifically, the values of “T” (tumor size) and “N” (lymph node invasion) would only increase in accuracy after pathological staging (replacing any image- or exam-based clinical staging prior to any surgical event). All pathology reports included in TCGA are inherently related to the surgical excision of tumor; please see the metadata documentation for the TCGA dataset (“Pathologic Stage: Pathologic staging combines the results of both the clinical staging (physical exam, imaging test), see Clinical stage, with surgical results.” <https://docs.cancer-genomicscloud.org/docs/tcga-metadata>)

To clarify this point, we have made the following revision (see Methods, 2nd paragraph):

The TNM staging attribute used in this study is “pathological stage,” i.e., stage based on pathologist assessment of patient tumor slide(s) combined with previous clinical results. This value is distinct from “clinical stage” as provided by TCGA; we chose pathological rather than clinical staging for “ground truth” as (1) it is considered the diagnostic gold standard during the course of patient care and (2) information

concerning pathologic staging is contained within report text. Staging was determined in a systematic manner by TCGA across all patients [12]. All data used for ground truth labeling was derived from the TCGA metadata as provided by the TCGA data portal.

Reviewer Comment: “The methodology of applying a BERT-variant for classification is common practice in the field. There isn't enough details provided for this work to be reproduced. For example, to reproduce the results, one would need learning rate, number of training epoch, and how the 9,523 patients were selected. It is also unclear how the one pathology report per patient was selected. It is not uncommon to have more than one pathology report per cancer patient.”

Thank you for your comments. All model parameters, including learning rate and number of training epochs, can be found on the Github repository associated with this paper, <https://github.com/tatonetti-lab/tnm-stage-classifier>. This repository is referenced in “Code Availability” section. For further detail on patient and report selection, we refer to our previous paper, <http://dx.doi.org/10.2139/ssrn.4418621>, which describes how the patients were selected from the TCGA dataset as well as how 1 report/patient was selected. In the Methods section of the current paper, we describe the patient and report selection process for the CUIMC dataset in detail as well (see the first 2 paragraphs of the section “Characterization of CUIMC Pathology Report Dataset” in Methods).

Reviewers' Comments:

Reviewer #1:

Remarks to the Author:

All my concerns have been addressed. Thank you!

Reviewer #3:

Remarks to the Author:

Very interesting work and very timely. Some of the reviewer comments have been addressed. However, there are two remaining major concerns

1) Previous research in this area is insufficiently cited and discussed. Overall there are very few references. For example this study (and several others) have investigated the exact same problem and they need to be contrasted with the current study
<https://pathsocjournals.onlinelibrary.wiley.com/doi/pdf/10.1002/path.6232>

2) the methods are still not entirely up to date, and the response to the reviewer in this regard is not sufficient. ClinicalBERT and Clinical-BigBird are NOT SOTA LLMs. SOTA LLMs are listed on the Hugging Face leaderboard, e.g. Llama, Mixtral etc. At least one of those should be compared to the proposed approach. Otherwise the study will be outdated on the day of publication.

Response to Reviewer Comments

We appreciate the opportunity to revise our manuscript in response to the remaining reviewer comments. Those critiques (black text) and our response (blue text) and revision are below.

1) Previous research in this area is insufficiently cited and discussed. Overall there are very few references. For example this study (and several others) have investigated the exact same problem and they need to be contrasted with the current study
<https://pathsocjournals.onlinelibrary.wiley.com/doi/pdf/10.1002/path.6232>

We appreciate the need for a more complete discussion of the literature in this area. In addition to the provided study, we found other important references to include in our introduction. As a result, we have significantly revised the introduction to cover the use of generative AI for pathology report information extraction and added the following references:

Truhn, Daniel et al. "Extracting structured information from unstructured histopathology reports using generative pre-trained transformer 4 (GPT-4)." *The Journal of pathology* vol. 262,3 (2024): 310-319. doi:10.1002/path.6232

Choi HS, Song JY, Shin KH, Chang JH, Jang BS. Developing prompts from large language model for extracting clinical information from pathology and ultrasound reports in breast cancer. *Radiat Oncol J.* 2023 Sep;41(3):209-216. doi: 10.3857/roj.2023.00633. Epub 2023 Sep 21. PMID: 37793630; PMCID: PMC10556835.

Huang, J., Yang, D.M., Rong, R. et al. A critical assessment of using ChatGPT for extracting structured data from clinical notes. *npj Digit. Med.* 7, 106 (2024).

Beyond Self-Consistency: Ensemble Reasoning Boosts Consistency and Accuracy of LLMs in Cancer Staging 2024, 2404.13149, arXiv

2) the methods are still not entirely up to date, and the response to the reviewer in this regard is not sufficient. ClinicalBERT and Clinical-BigBird are NOT SOTA LLMs. SOTA LLMs are listed on the Hugging Face leaderboard, e.g. Llama, Mixtral etc. At least one of those should be compared to the proposed approach. Otherwise the study will be outdated on the day of publication.

We thank the reviewer for the comment and agree that BERT-based models are not state-of-the-art when considering the general purpose benchmarks that are used to

evaluate LLMs. However, BERT-based models do offer some advantages over more recent generative models, like Llama and Mixtral, in that they are efficient in terms of memory and computational time which makes them easy to fine-tune for tasks. Further, since they are not generative, the outputs have less variability and there is no risk of hallucination. Even so, we agree that a comparison of performance is relevant. We repeated the analysis using Llama 3. We found that the trained BERT models outperformed the base Llama 3 model and that BERT models outperformed a fine-tuned Llama 3 model in 2/3 tasks. We have revised the manuscript to include this additional analysis and results. Revisions were made to the introduction, results, methods, and discussion to reflect this new analysis.

Reviewers' Comments:

Reviewer #3:

Remarks to the Author:

Thank you for addressing my comments.